# RNA-seq and Mitochondrial DNA Analysis of Adrenal Gland Metastatic Tissue in a Patient with Renal Cell Carcinoma

**DOI:** 10.3390/biology11040589

**Published:** 2022-04-13

**Authors:** Tomoyoshi Komiyama, Hakushi Kim, Masayuki Tanaka, Sanae Isaki, Keiko Yokoyama, Akira Miyajima, Hiroyuki Kobayashi

**Affiliations:** 1Department of Clinical Pharmacology, Tokai University School of Medicine, Isehara 259-1193, Kanagawa, Japan; hkobayas@is.icc.u-tokai.ac.jp; 2Department of Urology, Tokai University Hachioji Hospital, Tokyo 192-0032, Japan; 3Medical Science College Office, Tokai University, Isehara 259-1193, Kanagawa, Japan; matanaka@tokai-u.jp (M.T.); sanaei@tsc.u-tokai.ac.jp (S.I.); kekoyoko@tsc.u-tokai.ac.jp (K.Y.); 4Department of Urology, Tokai University School of Medicine, Isehara 259-1193, Kanagawa, Japan; akiram@tokai.ac.jp

**Keywords:** adrenal gland, renal cell carcinoma, metastasis, whole mtDNA region, RNA-seq

## Abstract

**Simple Summary:**

We performed mutation analysis of ribonucleic acid sequencing (RNA-seq) and whole mitochondrial genome data from tissues collected from metastatic lesions of a male patient in his 60s with metastatic renal cell carcinoma. We confirmed the common mutation sites of a mitochondrial gene containing the T3394Y, R11,807G, and G15,438R sites. Pathway analysis, using RNA-seq data, confirmed the common mutant pathway between renal cell carcinoma and metastatic adrenal carcinoma as cytokine–cytokine receptor (CCR) interaction. However, no common nuclear gene mutations were identified. The two similar sequences suggest that mtDNA-mutation-driven metastasis to the adrenal gland may affect CCR interactions and cause cancer cells to thrive.

**Abstract:**

This study aimed to clarify whether genetic mutations participate in renal cell carcinoma (RCC) metastasis to the adrenal gland (AG). Our study analyzed whole mitochondrial gene and ribonucleic acid sequencing (RNA-seq) data from a male patient in his 60s with metastatic RCC. We confirmed common mutation sites in the mitochondrial gene and carried out Kyoto Encyclopedia of Genes and Genomes (KEGG) analysis using RNA-seq data for RCC and adrenal carcinoma. Furthermore, we confirmed the common mutation sites of mitochondrial genes in which the T3394Y (p.H30Y) site transitioned from histidine (His.; H) to tyrosine (Tyr.; Y) in the NADH dehydrogenase subunit 1 (*ND1*) gene. The R11,807G (p.T350A) site transitioned from threonine (Thr.; T) to alanine (Ala.; A). Additionally, the G15,438R or A (p.G231D) site transitioned from glycine (Gly.; G) to aspartic acid (Asp.; D) in cytochrome b (CYTB). Furthermore, pathway analysis, using RNA-seq, confirmed the common mutant pathway between RCC and adrenal carcinoma as cytokine–cytokine receptor (CCR) interaction. Confirmation of the original mutation sites suggests that transfer to AG may be related to the CCR interaction. Thus, during metastasis to the AG, mitochondria DNA mutation may represent the initial origin of the metastasis, followed by the likely mutation of the nuclear genes.

## 1. Introduction

We performed genetic analysis to investigate metastasis, using cancer tissues that had metastasized from the kidneys (renal cell carcinoma; RCC) to the adrenal gland (AG). This study aimed to clarify whether genetic mutations play a role in metastasis from RCC to the AG. To this end, we have been conducting research on both nuclear genes and mitochondrial DNA (mtDNA). Previously, we reported that mitochondrial DNA mutations are involved in the development and metastasis of RCC [1,2]. The kidney is rich in blood vessels; therefore, RCC is said to often metastasize to places with abundant blood flow, such as the lungs, bones, brain, liver, and AG [3,4,5]. However, RCC may also spread to the lymph nodes where lymphatic fluid is collected. Therefore, in this study, we focused on the relationship between metastasis and genetic mutations based on metastatic AG tissue.

The AG is an organ approximately 2–3 cm in size, located above the kidney, as shown in Figure 1 [6]. As such, the kidney is the most likely organ from which metastasis to the AG originates. Indeed, studies have reported RCC metastasis to the AG; however, these studies did not consider potential genetic involvement. Molecular-targeted drugs, such as angiogenesis inhibitors and mTOR inhibitors, have been developed for metastatic RCC, since the era of immunotherapy with interferon [7]. Furthermore, the development of immune checkpoint inhibitors has significantly improved survival rates.

Currently, drug treatments for metastatic RCC are diverse and stratified by risk factors using models such as those from the Memorial Sloan–Kettering Cancer Center and the International Metastatic RCC Database Consortium. However, it is difficult to select appropriate cases for study [8,9,10,11,12,13,14,15,16]. In addition, the effectiveness of therapeutic agents has been investigated primarily using nuclear gene information. However, for more appropriate case selection, it is necessary to not only collect information from the primary lesion of the metastatic RCC but also from molecules of the metastatic lesion. Furthermore, research on biomarkers specific to RCC metastasis, including various tumor mutations, has primarily included analysis of the nuclear genome. Meanwhile, the mutation burden and mutation profile of the mitochondrial genome has only recently been characterized. In fact, numerous genetic studies, including those identifying biomarkers related to the mechanism and treatment of RCC, have been conducted over the last 5 years [6,7,17,18,19,20,21,22,23,24,25,26,27,28,29].

Previously, our group analyzed the recurrence and survival of the D-loop and NADH dehydrogenase subunit 1 (*ND1*) regions of mitochondrial DNA in 62 patients [1,16,30,31]. We showed that adverse pathological tumor features in localized RCC are affected by high mutation rates in the mitochondrial D-loop region. Moreover, compared to patients with *MT-ND1* or D-loop mutations, we found that RCC patients with both D-loop and *MT-ND1* mutations exhibit more advanced cancer-specific survival. Thus, differences in nuclear gene expression and mitochondrial genome mutation profiles between primary and metastatic lesions may aid in the selection of various therapeutic agents. Therefore, we performed mutation analysis of RNA-seq and mitochondrial genome data using surgical specimens of tissues from patients with metastatic RCC collected from metastatic lesions. In the current study, we aimed to verify whether these genes are associated with metastasis. In addition, we sought to identify a more effective molecular-targeted drug for metastatic RCC by investigating pre-administration information, selected drugs, and their effects, as well as the prognosis relative to the administration method based on genomic analysis. We believe that these results provide valuable insights regarding the selection of effective drugs for the treatment of metastatic cancer, as well as the relationship between cancer-specific mutant genotypes and metastatic destinations.

## 2. Materials and Methods

### 2.1. Case Description, Timeline, and Clinical Intervention

A male patient in his 60s with type 1 diabetes mellitus was referred to our hospital in July 2017. Chest and abdominal computed tomography showed a 3.5 cm left renal mass with hypervascularity, a bilateral adrenal mass (left was 3 cm in diameter; right was 1.2 cm in diameter; Figure 1), multiple lung nodules, and bone metastasis (T3, 10; L 3, 4 vertebrae). The findings of the bone lesion biopsy were compatible with features of clear cell RCC. In August 2017, the patient underwent cytoreductive left nephrectomy with ipsilateral adrenalectomy, without neoadjuvant therapy. The pathological findings of the kidney tumor revealed clear RCC, with a maximum diameter of approximately 40 mm, Fuhrman grade 2 tumor, v0, ly0 (sample number: K58-R). Pathological examination of the left AG (sample number: K58-M) revealed metastatic cancer foci with a pseudocapsule. Both DNA and RNA were extracted from the normal kidney tissue (sample number: K58-N), the primary RCC site, and the metastatic site of the left AG.

Following the guidelines of the Declaration of Helsinki, this study was conducted with approval from the institutional review board (ethics committee) of Tokai University, Japan (protocol code 15R065; date of approval: 12 August 2015). Informed consent was obtained from the patient.

#### 2.1.1. DNA Extraction

The surgical specimens of the cancerous and normal kidney tissues were collected, immersed in RNAlater (Qiagen, Valencia, CA, USA), and stored at −80 °C. The cancerous and normal kidney tissues for study were then collected from the surgical specimens by macroscopic judgment. In addition, a blood sample was collected from the peripheral veins the day before surgery. DNA from blood sample was used as the control sequence.

#### 2.1.2. Extraction of DNA and RNA

Total DNA and RNA were extracted from cancer tissues using TRIzol (Invitrogen, Carlsbad, CA, USA), followed by RNeasy (Qiagen, Valencia, CA, USA) (Table 1). The RNA preparation was assessed for degradation using an Agilent 2100 Bioanalyzer (Agilent Technologies, Palo Alto, CA, USA).

#### 2.1.3. Extraction of Mitochondrial DNA

Tissue and blood samples were then incubated with 200 g of proteinase K in buffer (10 mM Tris-Cl (pH 8.0), 150 mM NaCl, 10 mM ethylenediaminetetraacetic acid (EDTA), 0.1% sodium dodecyl sulfate) at 60 °C overnight, and later extracted with phenol–chloroform (phenol–chloroform–isoamyl alcohol, 25:24:1) twice at 11,000 rpm for 10 min at room temperature (25 °C). DNA was precipitated by the addition of 0.1 volume of 3 M Na-AcOH and 2.5 volumes of ice-cold ethanol. Then, after centrifugation at 15,000 rpm at 4 °C for 20 min, the DNA pellets were rinsed with 70% cold ethanol, dried, and dissolved in TE buffer (10 mM Tris-HCl (pH 8.0) and 1 mM EDTA). The total mtDNA was then amplified by polymerase chain reaction (PCR) with the primer sets (Appendix A), which cover the whole mtDNA region [32,33].

### 2.2. PCR Amplification Conditions

PCR was performed using KOD FX Neo (Toyobo, Osaka, Japan) under the following conditions: melting at 98 °C for 5 s, annealing at 63–66 °C for 15 s, and then extension at 68 °C for 20 s, for a total of 35 cycles. The PCR products were treated with EXOSAP-IT (Ametrix, Santa Clara, CA, USA) and immediately sequenced using a Big Dye Terminator v3.1 Reaction Kit (Applied Biosystems, Torrance, CA, USA) and an ABI 3500xL DNA sequencer (Life Technologies, Carlsbad, CA, USA).

### 2.3. Sequencing Analysis of Mitochondrial DNA

Complete mtDNA sequence data from the patient with localized RCC were assembled using Sequencher 5.0 software (Gene Codes Corporation, Ann Arbor, MI, USA). The mutation sites were determined using NC_012920.1 as the reference sequence.

### 2.4. RNA-seq Analysis: Mapping and Expression Analysis

The raw reads were obtained from the HiSeq 2500, and after completion of the run, data were base-called and demultiplexed (provided as Illumina FASTQ 1.8 files, Phred+33 encoding). FASTQ format files in Illumina 1.8 format were considered for downstream analysis (Illumina, San Diego, CA, USA).

#### 2.4.1. Sequencing Libraries

Sequencing libraries were prepared using the TruSeq Stranded mRNA Library Prep Kit (Illumina), according to the manufacturer’s instructions [34,35].

#### 2.4.2. Sequencing (HiSeq2500 PE100)

The libraries were sequenced on a HiSeq 2500 sequencing platform using HiSeq SBS Kit v4 reagents with 2 × 101 cycles.

#### 2.4.3. Data Analysis (Human RNA-seq Tophat/Cufflinks)

The adapter sequences and low-quality regions were trimmed using Cutadapt (v1.1) (National Bioinformatics Infrastructure Sweden, Uppsala, Sweden) and Trimmomatic v0.32 (RWTH Aachen University, Aachen, Germany), respectively. After pre-processing, the reads were mapped to the human reference genome GRCh38 using TopHat (v2.0.14) (Johns Hopkins University, MD, USA) [36]. Differential gene expression analysis was performed using the Cufflinks package (v2.2.1)(University of Washington, Seattle, USA) [37,38].

### 2.5. RNA-seq Analysis and Gene Expression Analysis Based on DAVID

We created lists annotated to *Homo sapiens* genes by enrichment analysis. The listed genes were submitted to the DAVID functional annotation database (http://david.abcc.ncifcrf.gov/) (accessed on 7 December 2021), providing a broad and unguided test against primarily Gene Ontology (GO) groups as well as Kyoto Encyclopedia of Genes and Genomes (KEGG) pathways [30,33,39,40]. When GO gene groups and KEGG pathways were extracted, the expression levels increased four times. The results are considered statistically significant when the *p*-value is <0.05.

## 3. Results

### 3.1. Somatic Mutations in the Mitochondrial DNA Region

In two cancer tissues, a total of five site mutations were confirmed compared to those in normal cells (Table 2). The mutations occurred in the sites numbering 71; 1826; 3394; 11,807; and 15,438. Site 71 of the D-loop region changed to a gap (71delG). The G1826 site in normal cells was mutated to the G1826 R of 16S ribosomal RNA in both cancer cell lines. It was also found that the T3394C site of *ND1* in normal cells was mutated to the heterosite of T3394Y (Y: Pyrimidine) in cancer cells. Site 15,438 changed from G in cytochrome b (*CYTB*) to R or A. In addition, site 11,807 of the NADH dehydrogenase subunit 4 (*ND4*) gene changed to G from the R heterosite (Table 2).

Moreover, the G15,438R or A (p.G231D) site transitioned from glycine (Gly.; G) to aspartic acid (Asp.; D), while the R11,807G (p.T350A) site transitioned from threonine (Thr.; T) to alanine (Ala.; A). Furthermore, the T3394Y (p.H30Y) site transitioned from histidine (His.; H) to tyrosine (Tyr.; Y) (Table 3). However, the signal of the T3394Y thymine (T) was lower than that of cytosine (C). Hence, it was confirmed that K58-R and K58-M have similar sequences.

### 3.2. RNA-seq Analysis and Gene Expression Analysis Based on DAVID

The KEGG pathways in which the expression and activity levels of adrenal cancer were more than doubled were primarily enriched in “Complement and coagulation cascades”, “Metabolic pathways”, and “Cytokine–cytokine receptor interaction” (Table 4). Additionally, pathways in which the expression and activity levels were more than doubled in RCC were primarily related to “Cell adhesion molecules (CAMs)”, “Antigen processing and presentation”, “Neuroactive ligand–receptor interaction”, “Cytokine”, and “cytokine receptor interaction” (Table 5). In particular, the cytokine–cytokine receptor interaction is of interest in this pathway. The expression level of this KEGG pathway was more than doubled in both cancer cell lines. However, there was a clear difference in the CXC subfamily pathways. That is, in adrenal cancer cells, the pathways of eight CXC subfamilies (CXCR1: *CXCL1*, *5, 6,* and *8*; CXCR2: *CXCL2* and *3*; CXCR3: *CXCL4*; and *CXCL4L1*) were affected (Figure 2, Appendix A and Appendix A). Meanwhile, in RCC, differences were observed in the expression levels of the four genes for CXCR3, which interacts with *CXCL9*, *CXCL10*, *CXCL11*, and *CXCL13* (Figure 2 and Appendix A). Furthermore, the TGF-β family (*BMPR1A* and *BMPR1B*) was also impacted (Appendix A).

### 3.3. Clinical Course after Surgery

After surgery, sunitinib was administered at a dose of 37.5 mg in September 2017. Owing to the progression of the lumbar vertebral metastasis, the patient was administered nivolumab as second-line systemic therapy. Although the metastatic lesions showed a partial response to nivolumab, the agent was discontinued due to hypopituitary insufficiency (central adrenal insufficiency), and oral intake of steroids was initiated. Radiotherapy was administered for the bone metastases. As of October 2021, multiple lung metastases and a right adrenal metastasis-maintained shrinkage, as well as multiple bone metastases, were shown in osteosclerotic images.

## 4. Discussion

This study aimed to clarify the involvement of genetic mutations in the metastasis of RCC to the AG. We analyzed the total mitochondrial DNA and RNA sequences of the renal and adrenal cancer tissues in a patient with RCC metastasis to the AG. By analyzing mitochondrial and nuclear DNA from two cancer sequences (K58-R and K58-M), we were able to confirm whether a common gene for cancer metastasis exists. Specifically, three common mutation sites were identified: T3394Y, R11,807G, and G15,438. T3394C mutation is reportedly associated with colon cancer and leukemia [41,42]. However, it likely became heteroplasmic when it transitioned into a cancer cell type in this case study. Our previous studies have reported the involvement of mtDNA mutations in cancer metastasis [1,2]. Therefore, since they share a common mitochondrial sequence, it is considered that the mutation at this site had metastasized from the primary RCC to the AG. In addition, the other two mutations have not been confirmed in other cancer reports.

Regarding *ND1*, Hayashi et al. showed that mtDNA mutations are involved in cancer metastasis by creating cybrid cells via cytoplasmic transplantation using mouse cells [43]. Complex I is a mitochondrial respiratory enzyme complex that is the main source of reactive oxygen species (ROS) [44]. This confirmed that the occurrence of mitochondrial mutations induces a decrease in mitochondrial respiratory activity and an abnormality in glycolytic activity from the causal relationship of ROS production, leading to carcinogenesis. Thus, it is highly likely that this mutation in *ND1* had metastasized from the primary RCC to the AG. Mitochondria are likely to be involved in adrenal metastasis, as RCC developed earlier in this patient. Furthermore, in our report, there was a correlation between the recurrence of *ND1* and RCC (61 patients) and cancer-related death. These data are provided as part of our supplementary data.

Moreover, in the pathway analysis based on RNA-seq, the common mutant pathway cytokine–cytokine receptor interaction between RCC and adrenal carcinoma was confirmed. In particular, it strongly affected *CXCL1-11* (excluding *CXCL7*), as well as 13 genes of the CXC subfamily. It was also determined that the CXC subfamily that comprises the pathway is affected by three genes: *CXCR1*, *CXCR2*, and *CXCR3*. Zeng et al. found that the transcriptional levels of *CXCL1/2/5/6/9/10/11/16* in RCC tissues were reduced, while those of *CXCL3/7/12/13* were elevated. *CXC1/5/9/10/11/13* were shown to have a significant correlation with the pathological stage of RCC. RCC patients with low transcriptional levels of *CXCL1/2/3/5/13* have a better prognosis [45]. Oldham et al. showed that CCR5, CXCR3, and CXCR6 are involved in selective recruitment of T cells into RCC tissues. These chemokine receptors, as well as CCR6, are involved in recruiting Tregs to the tumor site [46]. Furthermore, in RCC, in addition to the *CXCR3* gene, the transforming growth factor (TGF)-β family (BMPR1A, BMPR1B, ACVR1) pathway is also impacted. Specifically, the expression levels of bone morphogenetic protein (BMP) receptors Acvr1, Bmpr1a, and Bmpr1b in the BMP signal transduction system were decreased. BMP signaling can promote tumor growth and suppress endometrial cancer. Fukuda et al. demonstrated the tumor-promoting effects of BMP signaling in endometrial cancer cells by inducing cancer stemness, EMT, and migration [47]. Several secreted proteins, including TWSG1 and Gremlin, antagonize BMP ligands [47].

TGF-β is a major inducer of the epithelial–mesenchymal transition (EMT) [48,49]. The TGF-β pathway is reportedly activated in the endothelial cells (ECs) [50,51,52], and TGF-β has tumor-promoting and tumor-suppressing effects in the EC. However, the effects of BMPs, members of the TGF-β family, on ECs are not well-known [53,54,55]. Complexes of type I and type II serine/threonine kinase receptors are induced by the cellular effects of BMP ligands [56,57,58]. ACVRL1 (ALK1), ACVR1 (ALK2), BMPR1A (ALK3), and BMPR1B (ALK6) are classified as type I receptors, and ACVR2A (ActRII), ACVR2B (ActRIIB), and BMPR2 (BMPRII) as type II receptors [47,59]. Following receptor activation, SMAD1/5/8 are phosphorylated and form complexes with SMAD4. They are translocated to the nucleus, where they regulate the transcription of several target genes, including *ID1* [60,61,62,63,64,65,66]. In fact, we did not find the same gene or mutation site in our RNA-seq and mtDNA analyses using leukemia MOLT-3 cells. A mutation in mitochondrial DNA occurs in the background of metastasis. Subsequently, the mutated mitochondrial DNA is transferred to normal cells by interaction, and the pathway is affected by the mutation of each gene.

Molecular-targeted drugs are used to treat metastatic RCC. Improvements in survival rates have been reportedly compared, and treatment strategies for metastatic RCC are undergoing major changes. However, appropriate risk assessment, prognosis prediction, and optimal administration of molecular-targeted drugs for metastatic RCC in Japan have not yet been established. Therefore, it is necessary to consider information on a large number of patients to establish an effective treatment strategy.

## 5. Conclusions

Based on mitochondrial gene analyses, we confirmed the common mutation sites between the original RCC and metastatic adrenal carcinoma. The three sites, G15,438R, R11,807G, and T3394Y, helped us confirm that K58-R and K58-M have similar sequences. Moreover, a common mutant pathway, CCR, was identified between RCC and adrenal carcinoma. However, no common nuclear gene mutations were identified. Furthermore, in this study, we were not able to identify specific genes associated with metastasis. Nevertheless, the two similar sequences suggest that mtDNA-mutation-driven metastasis to the AG may affect CCR interactions, allowing cancer cells to thrive.

## Figures and Tables

**Figure 1 biology-11-00589-f001:**
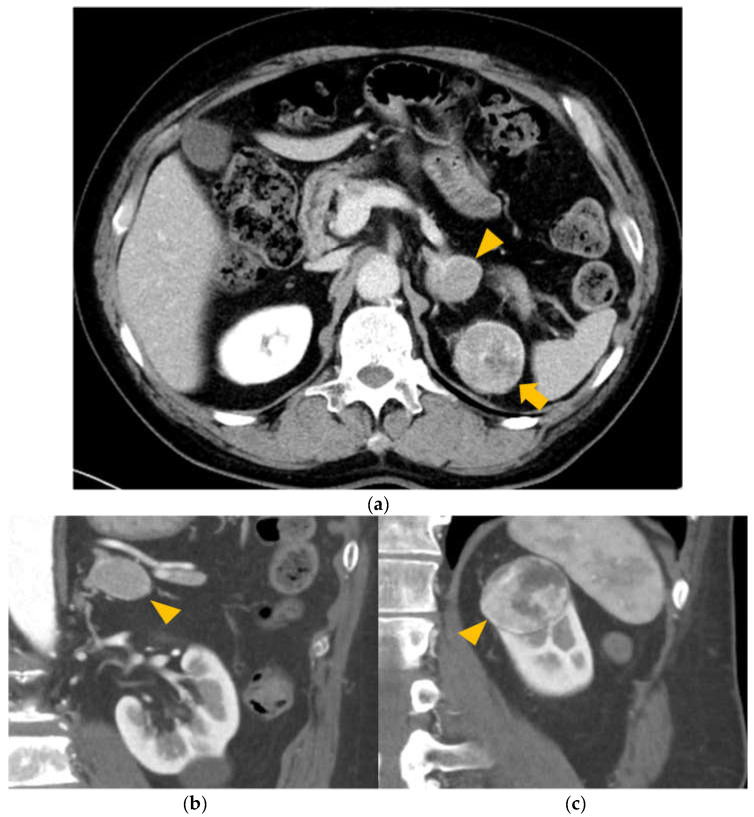
Position of metastatic and primary cancers. (**a**) Primary tumor in the left kidney (arrow) and metastatic tumor in the left adrenal gland (arrowhead). (**b**) Metastatic tumor in the left adrenal gland (arrowhead). (**c**) Renal cell carcinoma in the upper pole of the left kidney (arrowhead).

**Figure 2 biology-11-00589-f002:**
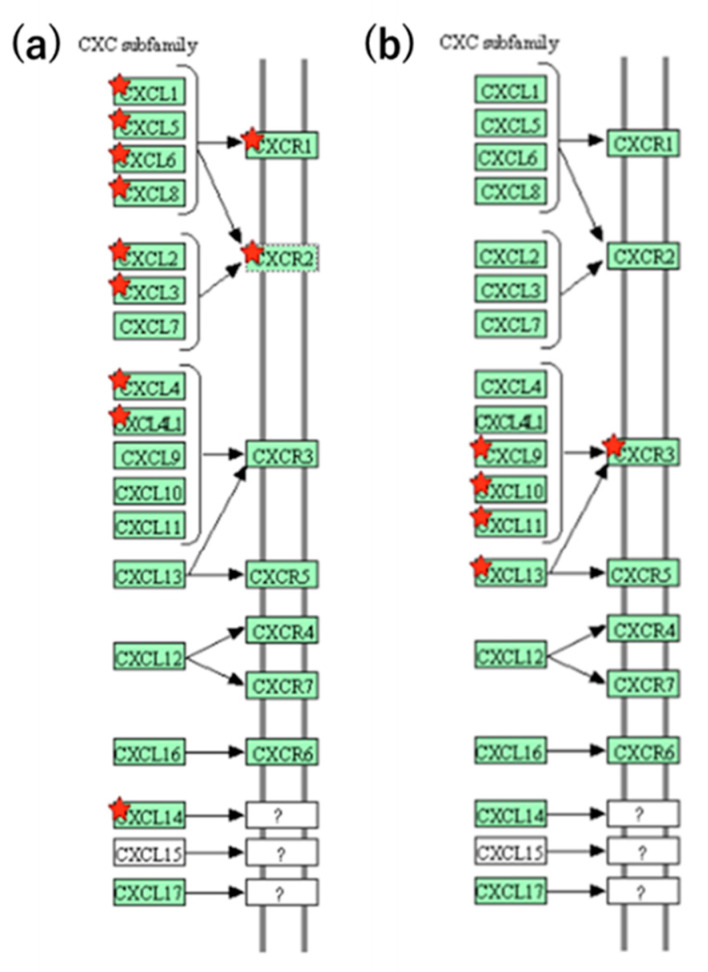
Association of CXC subfamily pathways in renal cell carcinoma (adrenal cancer cells). (**a**) Adrenal cancer cells. (**b**) Renal cell carcinoma. Red star: Expression-variable genes on the pathway.

**Table 1 biology-11-00589-t001:** Results of the RNA sample quality test.

No.	Sample Name	Nanodrop			Bioanalyzer		
		A260/A280	A260/A230	ng/μL	28S/18S	RIN	ng/μL
1	RCC: K58-R	2.00	2.14	3991.2	1.2	9.2	3088.1
2	AG: K58-M	1.99	2.06	3298.0	1.5	9.4	2692.7

**Table 2 biology-11-00589-t002:** Mitochondrial DNA mutation sites of three samples.

Position	Reference	Normal Cells	Cancer Cells	Metastatic Cells	Gene Name	dbSNP (Build 154v2)
	NC_012920	K58-N	K58-R	K58-M		
71	G	G	-	-		
73	A	G	G	G		rs869183622
153	A	G	G	G		rs370716192
263	A	G	G	G		rs2853515
315.1	-	C	C	C		
489	T	C	C	C		rs28625645
750	A	G	G	G		rs2853518
1041	A	G	G	G		rs58327546
1438	A	G	G	G		rs2001030
1826	G	G	R	R		
2706	A	G	G	G		rs2854128
3394	T	C	Y	Y	*ND1*	rs41460449
4491	G	A	A	A	*ND2*	rs201172504
4769	A	G	G	G	*ND2*	rs3021086
5951	A	G	G	G	*COX1*	rs7340122
7028	C	T	T	T	*COX1*	rs2015062
8701	A	G	G	G	*ATP6*	rs2000975
8860	A	G	G	G	*ATP6*	rs2001031
9115	A	G	G	G	*ATP6*	rs1603222091
9242	A	G	G	G	*COX3*	rs1603222192
9540	T	C	C	C	*COX3*	rs2248727
10,398	A	G	G	G	*ND3*	rs2853826
10,400	C	T	T	T	*ND3*	rs28358278
10,873	T	C	C	C	*ND4*	rs2857284
11,719	G	A	A	A	*ND4*	rs2853495
11,807	A	R	G	G	*ND4*	rs1603223419
12,705	C	T	T	T	*ND5*	rs193302956
13,434	A	G	G	G	*ND5*	rs1603224187
14,308	T	C	C	C	*ND6*	rs28357674
14,766	C	T	T	T	*CYTB*	rs193302980
14,783	T	C	C	C	*CYTB*	rs193302982
15,043	G	A	A	A	*CYTB*	rs193302985
15,301	G	A	A	A	*CYTB*	rs193302991
15,326	A	G	G	G	*CYTB*	rs2853508
15,438	G	G	R	A	*CYTB*	
16,223	C	T	T	T		rs2853513
16,234	C	T	T	T		rs368259300
16,300	A	G	G	G		rs879082592
16,316	A	G	G	G		rs1556424861
16,362	T	C	C	C		rs62581341
Total differences		38	41	41		

Y: pyrimidine (T or C), R: purine (G or A).

**Table 3 biology-11-00589-t003:** Amino acid substitution sites from the three sequenced samples.

Nucleotide Position	Reference			K58-N		K58-R		K58-M		Amino Acid Substitution	Gene Name
	NC_012920	Amino Acid Substitution Site/Abbreviation	1-Letter Abbreviation	Codon	1-Letter Abbreviation	Codon	1-Letter Abbreviation	Codon	1-Letter Abbreviation		
3394	TAT	30/Tyr	Y	CAT	H	YAT	H/Y	YAT	H/Y	p.H30Y	*ND1*
11,807	ACT	350/Thr	T	RCT	T/A	GCT	A	GCT	A	p.T350A	*ND4*
15,438	GGC	231/Gly	G	GGC	G	GRC	G/D	GAC	D	p.G231D	*CYTB*

**Table 4 biology-11-00589-t004:** K58-M using the Kyoto Encyclopedia of Genes and Genomes (KEGG) analysis; K58-M: fold change ≥2 upregulated 2243 genes (2152 genes*). * Number of genes corresponds to DAIVID.

Pathway Name	Count	% Count	*p*-Value	Benjamini
Complement and coagulation cascades	28	1.30111524	3.48 × 10^−11^	9.84 × 10^−^^9^
Metabolic pathways	158	7.34200743	1.07 × 10^−5^	0.00151811
Cytokine–cytokine receptor interaction	43	1.99814126	7.99 × 10^−5^	0.00753899
Glycine, serine, and threonine metabolism	13	0.60408922	1.70 × 10^−4^	0.01082157
Tryptophan metabolism	13	0.60408922	2.23 × 10^−4^	0.01082157
Retinol metabolism	17	0.78996283	2.29 × 10^−4^	0.01082157
Chemical carcinogenesis	19	0.88289963	3.93 × 10^−4^	0.0158922
Phenylalanine metabolism	8	0.37174721	5.59 × 10^−4^	0.01977878
Systemic lupus erythematosus	26	1.20817844	7.05 × 10^−4^	0.0221624
Steroid hormone biosynthesis	15	0.69702602	8.17 × 10^−4^	0.02310869
PPAR signaling pathway	16	0.74349442	0.00124254	0.03173786
Arachidonic acid metabolism	15	0.69702602	0.00138728	0.03173786
Drug metabolism—cytochrome P450	16	0.74349442	0.00145792	0.03173786

**Table 5 biology-11-00589-t005:** K58-R using the Kyoto Encyclopedia of Genes and Genomes (KEGG) analysis; K58-R: fold change ≥ 2 upregulated1897 genes (1810 genes*). * Number of genes corresponds to DAIVID.

Pathway Name	Count	% Count	*p*-Value	Benjamini
Cell adhesion molecules (CAMs)	28	1.54696133	1.18 × 10^−7^	3.02 × 10^−5^
Antigen processing and presentation	19	1.04972376	5.56 × 10^−7^	7.14 × 10^−5^
Type I diabetes mellitus	13	0.71823204	5.49 × 10^−6^	4.70 × 10^−4^
Cytokine–cytokine receptor interaction	34	1.87845304	1.38 × 10^−5^	7.76 × 10^−4^
Neuroactive ligand–receptor interaction	37	2.0441989	1.51 × 10^−5^	7.76 × 10^−4^
Graft-versus-host disease	11	0.60773481	1.90 × 10^−5^	8.14 × 10^−4^
Allograft rejection	11	0.60773481	5.71 × 10^−5^	0.00209757
Primary immunodeficiency	9	0.49723757	8.80 × 10^−4^	0.02827813
Calcium signaling pathway	23	1.27071823	0.00149072	0.0425683
Dilated cardiomyopathy	14	0.77348066	0.00181114	0.04654619

## Data Availability

The data presented in this study are available in the article and Appendix A.

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
