# Peer review of "RNA-seq and Mitochondrial DNA Analysis of Adrenal Gland Metastatic Tissue in a Patient with Renal Cell Carcinoma"

_biology, 2022, doi:10.3390/biology11040589_

Round 1
Reviewer 1 Report
In this study, Tomoyoshi Komiyama, etal,. described that “RNA-seq and Mitochondrial DNA Analysis of Adrenal Gland Metastatic Tissue in Patients with Renal Cell Carcinoma”. This is an interesting and well-done paper in a field of gene mutation compared between primary RCC tumor and metastatic AC tumor. The case study was carefully performed, and RNA-seq data analyzed well between RCC and AC. The experiments were carefully designed and performed. The data were well organized, and the text was well written. Author clearly analyzed the mutation in nuclear and mitochondria gene, there is no common nuclear gene mutation was seen between RCC and AC in the manuscript, interestingly showed in mitochondria gene mutation, these case study will understand the importance of mitochondria gene mutation has a role in cancer metastasis. I am very happy to recommend this manuscript for publication without any modification.Author Response
Dear Reviewer 1,
We would like to thank you for your careful review of our manuscript. We have addressed each one of your comments and have provided our responses as is elaborated below.
Sincerely,
Tomoyoshi Komiyama
Reviewer 1
In this study, Tomoyoshi Komiyama, etal,. described that “RNA-seq and Mitochondrial DNA Analysis of Adrenal Gland Metastatic Tissue in Patients with Renal Cell Carcinoma”. This is an interesting and well-done paper in a field of gene mutation compared between primary RCC tumor and metastatic AC tumor. The case study was carefully performed, and RNA-seq data analyzed well between RCC and AC. The experiments were carefully designed and performed. The data were well organized, and the text was well written. Author clearly analyzed the mutation in nuclear and mitochondria gene, there is no common nuclear gene mutation was seen between RCC and AC in the manuscript, interestingly showed in mitochondria gene mutation, these case study will understand the importance of mitochondria gene mutation has a role in cancer metastasis. I am very happy to recommend this manuscript for publication without any modification.
Reply: We sincerely thank you for your positive comment on our manuscript.
Reviewer 2 Report
This manuscript aimed to clarify whether genetic mutations are involved in the metastasis of renal cell carcinoma to the adrenal gland. The study analysed whole mitochondrial gene and ribonucleic acid sequencing (RNA-seq) data from a 60-year-old male patient with metastatic renal cell carcinoma. I believe that this manuscript may arouse the interest of some researchers and therefore propose to accept it.
Author Response
Reviewer 2
We would like to thank you for your careful review of our manuscript. We have addressed your comment and have provided our response as follows.
Sincerely,
Tomoyoshi Komiyama
This manuscript aimed to clarify whether genetic mutations are involved in the metastasis of renal cell carcinoma to the adrenal gland. The study analysed whole mitochondrial gene and ribonucleic acid sequencing (RNA-seq) data from a 60-year-old male patient with metastatic renal cell carcinoma. I believe that this manuscript may arouse the interest of some researchers and therefore propose to accept it.
Reply: We sincerely thank you for your positive comment on our manuscript.
Reviewer 3 Report
Manuscript is written well and discussion covered all the related aspects. Manuscript may be accepted. I have one suggestion on database repository. Is the gene expression data submitted to public data repository of Gene Expression Omnibus database?
Author Response
Reviewer 3
We would like to thank you for your careful review of our manuscript. We have addressed your comment and have provided our response as follows.
Sincerely,
Tomoyoshi Komiyama
Manuscript is written well and discussion covered all the related aspects. Manuscript may be accepted. I have one suggestion on database repository. Is the gene expression data submitted to public data repository of Gene Expression Omnibus database?
Reply: We thank you for your positive feedback on our manuscript. Following your instructions, we are registering our gene expression data to DDBJ.
Reviewer 4 Report
Manuscript entitled "RNA-seq and Mitochondrial DNA Analysis of Adrenal Gland Metastatic Tissue in Patients with Renal Cell Carcinoma" This work performed RNA-seq and mitochondrial DNA analysis in a RCC and its adrenal gland metastasis. There are some defects:
- The report is very preliminary which involve only one case. Accordingly no conclusion should be made.
- There is no nuclear DNA sequencing performed. So the authors should not claim that mitochondria DNA mutation may represent the initial origin of the metastasis, followed by the likely mutation of the nuclear genes.
- The rationale of this work and the clinical relevance is low. I can not understand the translational significance of this work. and why adrenal gland?
- Overall, the authors should perform DNAseq for nuclear genes. They should also include more cases to make a conclusion.
Author Response
Reviewer 4
We would like to thank you for your careful review of our manuscript. We have addressed each one of your comments and have provided our responses as follows.
Sincerely,
Tomoyoshi Komiyama
Manuscript entitled "RNA-seq and Mitochondrial DNA Analysis of Adrenal Gland Metastatic Tissue in Patients with Renal Cell Carcinoma" This work performed RNA-seq and mitochondrial DNA analysis in a RCC and its adrenal gland metastasis. There are some defects:
- The report is very preliminary which involve only one case.
Accordingly no conclusion should be made.
Reply: Thank you for your insightful suggestion. Since the Hayashi group reported that mitochondria are involved in cancer metastasis, we believe this data to be the first report that could be confirmed in human cancer patients. We considered the possibilities as a case report.
- There is no nuclear DNA sequencing performed. So the authors should not claim that mitochondria DNA mutation may represent the initial origin of the metastasis, followed by the likely mutation of the nuclear genes.
Reply: Analysis of mitochondrial genome confirmed the presence of the same mutation in both adrenal cancer tissue, metastasized from renal cell carcinoma, and the renal cell cancer tissue. It was different from the control. Since the Hayashi group also reported that mitochondria are involved in cancer metastasis, we believe this data to be the first report that could be confirmed in human cancer patients. It is also a very interesting fact that a common pathway mutation was confirmed by RNA analysis. Although, as rightly pointed out by the reviewer, DNA analysis has not been performed, we plan to perform MiSeq next-generation sequence analysis of 112 patients based on the results of future pathways. We plan to perform MiSeq next-generation sequence analysis, focusing on the genes related to the CXC subfamily and the associated pathway indicated by the RNA analysis performed in this study.
- The rationale of this work and the clinical relevance is low.
I can not understand the translational significance of this work. and why adrenal gland?
Reply: In this work, we mainly investigated the role of genes in cancer metastasis and focused on their extraction and sequencing.
For the purpose of this report, we were able to get fresh tissue samples of renal cell carcinoma and metastasized adrenal cancer. Since the kidney is the organ closest to the adrenal gland, and hence, most likely to be the source of metastasizing cells, we tried to extract candidate genes for metastasis by comparing these two tissue samples. The data from the mitochondrial DNA analysis yielded interesting results and while more investigation into the nuclear DNA of the samples is required, we believe that the comparison of carcinomatous tissue samples from these two adjacent organs provides valuable insights and it bears significance for future research.
- Overall, the authors should perform DNAseq for nuclear genes.
They should also include more cases to make a conclusion.
Reply: By using RNAseq in this gene analysis, we were able to derive the role of CXC subfamily, which is a common gene family involved in metastasis, and the associated pathways. In future, we will confirm the transfer of each gene described in the Discussion section and their roles by analyzing other tissue samples as well. The frequency of mutant sites will be investigated by MiSeq next-generation sequence analysis. Therefore, we will strive to increase the sample size and derive more data using the same method to improve the reliability of the analysis results.
5. English language and style are fine/minor spell check required
Reply:
We also revised our manuscript as the reviewer suggested.
We had our manuscript checked by a professional English editing service.
Round 2
Reviewer 4 Report
There is no significant improvement made.